# Phytochemical Properties of *Croton gratissimus* Burch (Lavender Croton) Herbal Tea and Its Protective Effect against Iron-Induced Oxidative Hepatic Injury

**DOI:** 10.3390/plants12162915

**Published:** 2023-08-10

**Authors:** Paul V. Ncume, Veronica F. Salau, Sibahle Mtshali, Kolawole A. Olofinsan, Ochuko L. Erukainure, Motlalepula G. Matsabisa

**Affiliations:** 1Department of Pharmacology, University of the Free State, Bloemfontein 9300, South Africa; paulvusi555@gmail.com (P.V.N.); veronica.salau@yahoo.com (V.F.S.); sbahlenosi105@gmail.com (S.M.); 2Laser Research Centre, Faculty of Health Sciences, University of Johannesburg, Doornfontein 2028, South Africa; kollyck@gmail.com (K.A.O.); loreks@yahoo.co.uk (O.L.E.)

**Keywords:** antioxidant, *Croton gratissimus*, in silico, phytochemicals, oxidative stress

## Abstract

Oxidative stress plays a vital role in the pathogenesis and progression of various liver diseases. Traditional medicinal herbs have been used worldwide for the treatment of chronic liver diseases due to their high phytochemical constituents. The present study investigated the phytochemical properties of *Croton gratissimus* (lavender croton) leaf herbal tea and its hepatoprotective effect on oxidative injury in Chang liver cells, using an in vitro and in silico approach. *C. gratissimus* herbal infusion was screened for total phenolic and total flavonoid contents as well as in vitro antioxidant capacity using ferric reducing antioxidant power (FRAP) and 2,2-diphenyl-1-picryl-hydrazyl (DPPH) methods. Oxidative hepatic injury was induced by incubating 0.007 M FeSO_4_ with Chang liver cells which has been initially incubated with or without different concentrations (15–240 μg/mL) of *C. gratissimus* infusion or the standard antioxidants (Gallic acid and ascorbic acid). *C. gratissimus* displayed significantly high scavenging activity and ferric reducing capacity following DPPH and FRAP assays, respectively. It had no cytotoxic effect on Chang liver cells. *C. gratissimus* also significantly elevated the level of hepatic reduced glutathione (GSH), superoxide dismutase (SOD), and catalase activities as well as suppressed the malondialdehyde (MDA) level in oxidative hepatic injury. Liquid Chromatography–Mass Spectrometry (LC-MS) analysis of the herbal tea revealed the presence of 8-prenylnaringenin, flavonol 3-O-D-galactoside, caffeine, spirasine I, hypericin, pheophorbide-a, and 4-methylumbelliferone glucuronide. In silico oral toxicity prediction of the identified phytochemicals revealed no potential hepatotoxicity. Molecular docking revealed potent molecular interactions of the phytochemicals with SOD and catalase. The results suggest the hepatoprotective and antioxidative potentials of *C. gratissimus* herbal tea against oxidative hepatic injury.

## 1. Introduction

The liver plays a vital role in regulating numerous metabolic and physiological functions in the body, as it is the primary site for the metabolism and detoxification of drugs, food, and other xenobiotics. Thus, the liver is often exposed to oxidative insult that results in complex liver diseases [1]. Several chronic liver diseases, including alcoholic liver disease, hepatic fibrosis, hepatic steatosis, hepatitis, and cirrhosis, are complex disorders causing morbidity and mortality, with approximately 2 million global mortality cases related to liver diseases [2,3].

Oxidative stress has been reported to play a pivotal role in the pathogenesis of liver diseases and facilitate their progression [2]. In the liver, the mitochondria and endoplasmic reticulum are the major sites of reactive oxygen species (ROS) generation. Excessive generation of ROS instigates liver damage as a result of alteration in hepatic proteins, lipids and DNA composition, and modulation of pathways that regulate normal biological processes. These events lead to the amplification of inflammatory responses, thereby resulting in the induction of necrosis and apoptosis of hepatocytes [2,4]. Alcohol consumption, high-calorie diet, drug overdose, and environmental pollutants are some of the factors implicated in the manifestation of liver injury via the generation of ROS [4]. Hepatotoxicity induced by heavy metals such as cadmium, copper, arsenic, and iron, is another cause of liver diseases [5]. Excess levels of hepatic iron have been implicated in the induction of oxidative stress and the progression of liver diseases such as non-alcoholic liver disease, steatohepatitis, viral hepatitis, and hemochromatosis. As the principal storage site of iron and synthesis of proteins involved in iron homeostasis, the liver is often highly susceptible to the pathological outcome of iron-induced oxidative damage [6]. Due to the critical role of oxidative stress in liver diseases, antioxidants are thus considered as a good therapeutic strategy to scavenge iron-induced ROS and treat liver disorders [4].

The use of medicinal plants in traditional medicine has continued to gain global interest owing to the presence of numerous phytochemicals such as polyphenols and flavonoids with strong antioxidant properties. Thus, several phytomedicines or herbal formulations are now commonly exploited for the prevention and treatment of numerous liver disorders, including hepatotoxicity [1]. One example of such plants is *Croton gratissimus. Croton gratissimus Burch,* also known as *Croton antunesii Pax*, *Croton zambesicus Mull. Arg,* lavender croton, or lavender fever berry, belongs to the family of *Euphorbiaceae*. The croton genus is a diverse and complex group of flowering plants which range from herbs to shrubs and trees [7]. These species of plants have been widely used in ethnomedicines in Africa due to its reservoir of phytochemicals [8]. The stem bark of *C. gratissimus* is widely used in different ethnical communities in South Africa for treating several ailments. Particularly, it is used by the Zulu communities of South Africa for managing stomach and intestinal disturbances owing to its purgative property [9]. Other ethnomedicinal uses of *C. gratissimus* include the treatment of arthritis, hypertension, gonorrhoea, diabetes, and malaria [10]. Studies have revealed the antioxidative effect of *C. gratissimus* against neurodegeneration and cholinergic dysfunction [11]. Its anti-inflammatory and anti-HIV properties have also been reported [12].

Despite the reported biological activities of *C. gratissimus,* there is a paucity of information on its effect on liver disorders. The present study was thus carried out to investigate the protective effect of *C. gratissimus* on oxidative hepatic injury induced by ferrous sulphate in Chang liver cells, by determining its effect on redox imbalance and proinflammation as well as identifying potential phytochemicals that may be responsible for its hepatoprotective property.

## 2. Results

The total phenolic content of the hot infusion of *C. gratissimus* was 28.9 ± 0.87 (mg/g/GAE), whereas its total flavonoid content was 102.24 ± 2.26 (mg/g/QE).

As represented in Figure 1A, *C. gratissimus* infusion significantly (*p* < 0.05) scavenged DPPH radical from the lowest to the highest concentration (15–240 μg/mL) with a low IC_50_ value of 0.01 μg/mL (Table 1). *C. gratissimus* demonstrated similar scavenging capabilities as those of the two standards, gallic acid with (0.02 μg/mL), and ascorbic acid (0.01 μg/mL). Additionally, the infusion exhibited an increasing ferric-reducing capability in a dose-dependent trend as those of the two standards, as depicted in Figure 1B. However, gallic acid (IC_50_: 42.12 μg/mL) outperformed ascorbic acid and the infusion.

*C. gratissimus* herbal infusion had no cytotoxic effect on Chang liver cells as there was no significant difference in cell viability between *C. gratissimus*-treated cells and normal Chang liver cells (Figure 2). However, compared to the normal control, cells treated with doxorubicin had significantly (*p* < 0.05) reduced viability.

As portrayed in Figure 3, induction of oxidative hepatic injury with iron sulphate significantly (*p* < 0.05) reduced the level of GSH in the Chang liver cells. However, pre-treatment with *C. gratissimus* significantly (*p* < 0.05) elevated GSH levels in a dose-dependent manner to levels comparable with the two standards, ascorbic acid and gallic acid.

Induction of oxidative hepatic injury led to a significant (*p* < 0.05) suppression in the activities of the superoxide dismutase (SOD) enzyme in the Chang liver cells as depicted in Figure 4. Incubation of the cells with the infusion led to a significant increase in the activities of SOD, which is slightly dose-dependent and similar to the activities of ascorbic acid and gallic acid.

As shown in Figure 5, there was a notable reduction in the activity of the catalase on the induction of oxidative injury in Chang liver cells. The reduced catalase activity was increased in cells pre-treated with *C. gratissimus* infusion in a dose-dependent manner. However, there was no significant difference in the catalase activities of the untreated cells and cells pre-treated with the infusion at the lowest concentration (15 μg/mL). Ascorbic acid displayed the best catalase activity.

As depicted in Figure 6, the untreated cells exhibited significantly (*p* < 0.05) higher MDA levels when compared to the normal control, except at the lowest concentration (15 μg/mL). The cells pre-treated with various concentrations of *C. gratissimus* significantly (*p* < 0.05) decreased the MDA levels in the cells and were similar to the control level at the highest concentration (240 µg/mL). The effect of *C. gratissimus* on the hepatic MDA levels compared favorably with the standards (ascorbic acid and gallic acid).

Cells incubated with iron sulphate had significantly (*p* < 0.05) increased levels of nitric oxide when compared with the normal as shown in Figure 7. Pre-treatment with *C. gratissimus* infusion significantly (*p* < 0.05) reduced nitric oxide levels to a level similar to the normal at the highest concentration. The activity of *C. gratissimus* at all the concentrations competed favorably with the two standards.

LC-MS analysis of the secondary metabolites present in the tea infusion, as presented in Table 2, revealed the presence of 8-prenylnaringenin and flavonol 3-O-D-galactoside flavonoid derivatives, as well as alkaloid molecules including caffeine and spirasine I. Additionally, other natural products, namely hypericin, pheophorbide-a, and 4-methylumbelliferone glucuronide, that can elicit various pharmacological activities were found in the *C. gratissimus* infusion. One limitation of this study is that we could not quantify the identified compounds of *C. gratissimus* infusion. However, qualitative analysis and compound isolation studies are being considered for future studies.

The oral toxicity in silico prediction result for *C. gratissimus* tea phytoconstituents, as displayed in Table 3, indicates that they do not possess any potential hepatotoxicity or adverse reaction towards Nrf2/ARE complex formation, which is vital for antioxidant protein gene expression. Furthermore, the LD_50_ of the compounds ranges between 40–5000 mg/kg and their toxicity class is from 2 to 5 according to the Globally Harmonized System of Classification (GHS) of labeling chemicals.

The molecular docking data revealed that the LC-MS identified tea phytochemicals exhibited negative binding affinities with selected antioxidant enzymes (Table 3). While Pheophorbide-a with the lowest binding energy (−12.6 Kcal/mol) exhibited the strongest affinity with catalase, 4-methylumbelliferone glucuronide with −7.6 Kcal/mol binding energy exhibited the strongest affinity with SOD. According to the 2D images in Figure 8, Pheophorbide-a interacted via hydrogen bonding with ARG 49, ILE 309, and HIS 339 amino acids at the catalase active site. Other forces that could have contributed to the phytochemical’s strong affinity for the enzyme include Pi-sigma, Pi-Pi stacked, Pi-cation, and Pi-alkyl bonds formed with other amino residues of the protein. However, in Figure 9, two favorable interactions including a hydrogen and pi-alkyl bond with ILE 113, ILE 266, and ILE 304 at the catalytic pocket of SOD may be attributed to the 4-methylumbelliferone glucuronide firm affinity for the enzyme.

The predicted pharmacokinetics of *C. gratissimus* tea compounds, as shown in Table 4, suggest the high gastrointestinal absorption (GIA) properties of 8-prenylnaringenin, spirasine I, flavonol 3-O-D-galactoside, and caffeine. Although pheophorbide-a, 4-methylumbelliferone glucuronide, and hypericin were predicted to have low GIA, they do not possess potential inhibitory effects on cytochrome (CYP1A2, CYP2D6, CYP3A4) enzymes vital in liver drug metabolism. Moreover, besides hypericin which has double Lipinski and lead likeness violations, most of the other compounds in the tea had only one or no violations of these pharmacokinetic indices, which also include Veber and PAINS alert descriptors.

## 3. Discussion

Hepatic oxidative damage contributes significantly to hepatotoxicity and the development and progression of several acute and chronic liver diseases that can instigate high morbidity and mortality, if not properly managed [2,13]. Traditional medicines have been widely studied for decades for the purpose of treating various human disease conditions including liver diseases. This is due to the presence of phytochemicals with strong antioxidant properties [14]. *C. gratissimus* herbal tea was investigated in the current study for its hepatoprotective effect against oxidative hepatic injury.

Oxidative stress is a principal determinant in the pathogenesis and progression of several liver diseases, hence the use of medicinal plants or herbal formulations with natural antioxidant properties to treat and manage these diseases has continued to increase. This characteristic has been attributed to the presence of bioactive compounds such as the phenolics and flavonoids [1]. The *Croton* genus has been studied for years, and several active compounds such as flavonoids, phenolics, diterpenoids, and sesterterpenoids that have the potential to act as antioxidants have been isolated and identified [15]. The high phenolic and flavonoid contents displayed by C. *gratissimus* in the present study suggest that the plant has potent antioxidant potential. This is consistent with the findings of Mfotie Njoya, Eloff, and McGaw [7], who reported high levels of phenolic and flavonoid contents in various extracts of C. *gratissimus.* The antioxidant effect of phenolics and flavonoids compounds are attributed to their capability of donating delocalized π electrons from their ring structures to scavenge free radicals [16], chelate metal ions such as Cu^2+^ and Fe^2+^, act as enzyme inhibitors, reducing agents, and singlet oxygen quenchers [17]. In accordance, the potent antioxidant properties portrayed by *C. gratissimus* (Figure 1A,B) may indicate the strong antioxidant capacity of the plant. These results are in line with the study of Ndhlala et al. [11] who reported the in vitro antioxidative activities of the leaf extracts and isolated compounds of two croton species, *C. gratissimus* and *C. zambesicus.*

To further support the in vitro antioxidant capacity of *C. gratissimus*, the antioxidant potential of the plant was carried out on oxidative injury in Chang liver cells. GSH constitute the highest cellular non-protein thiol and the most important cellular redox status determinant, with their highest abundance located in the liver [18]. A high GSH level is required to protect the liver from oxidative stress-mediated liver diseases since exposure to alcohol, environmental pollutants, and drugs, amongst other factors, can generate ROS and cause a shift in the hepatic cellular redox status. Thus, maintenance of GSH homeostasis is critical for hepatoprotection [18,19]. In the present study, the reduced GSH levels in untreated Chang liver cells (Figure 3) implied the occurrence of oxidative stress, as reduced GSH levels have been regarded as an oxidative stress marker at the cellular level [20]. Elevated cellular levels of GSH significantly reduce oxidative stress and cellular toxicity [21]. Thus, the increased level of GSH in cells pretreated with *C. gratissimus* (Figure 3) may suggest the hepatoprotective property of the herbal tea.

A principal mechanism of iron toxicity is the excessive generation of superoxide (O_2_·^−^) and hydroxyl (OH·) radicals [20]. The SOD enzyme is an active oxygen radical scavenger and a first line of defense for organisms during oxidative attack or exposure to toxicity. In hepatic tissue, the level of SOD activity can be used in assessing the degree of liver injury, as depleted concentration of SOD or surplus generation of O_2_·^−^ can result in oxidative stress. Thus, regulation and augmentation of SOD is pivotal in maintaining hepatic antioxidant integrity [22]. During Fe^2+^ oxidation, O_2_·^−^ is excessively generated and SOD dismutates O_2_·^−^ to H_2_O_2_ and O_2_. H_2_O_2_ is further broken down to H_2_O by catalase [23]. The remaining H_2_O_2_ not broken down to H_2_O by catalase reacts with Fe^3+^ to produce OH·that attack membrane lipids and initiate lipid peroxidation [20,24]. The extent of membrane lipid peroxidation can be determined by the concentration of MDA which can further be used to measure the degree of oxidative damage in a tissue. MDA is on one of the sensitive biomarkers of hepatic injury [22,25]. In the present study, the depleted activities of SOD (Figure 4) and catalase (Figure 5), as well as an increased level of MDA (Figure 6) in the untreated cells, may indicate high concentrations of O_2_·^−^ and H_2_O_2_, and increased lipid peroxidation, respectively. Additionally, excess O_2_·^−^ can react with nitric oxide (NO) to produce a highly reactive peroxynitrite (ONOO^-^) radical, a well-known mediator of proinflammation [26,27]. The elevated level of NO (Figure 7) in the untreated cells with the observed depleted SOD activities (Figure 4), could result in increased ONOO^-^ radical generation. Consequently, the improved activities of SOD and catalase as well as reduced MDA and NO levels in cells pre-treated with the tea extract portray an anti-oxidative and anti-proinflammatory effect of the plant. These could be due to the molecular interaction of the identified phytochemicals (Figure 8 and Figure 9) and the enzymes’ active site amino residues (Table 3), or enhanced radical scavenging capacity as depicted in Figure 1A. Studies have indicated that conversion of O_2_·^−^ to H_2_O_2_ by SOD is a multistep process which is required for proper orientation of the active site amino acids for easy accessibility of the substrate with the inner catalytic cavity of the enzyme [28].

The cytotoxicity study is a critical step in the development of pharmaceutical agents. This process which helps to ensure the safe use of chemical substances for therapeutic purposes involves analyzing their cytotoxicity on normal cell lines [29]. In the present study, *C. gratissimus* showed no cytotoxic effect on Chang liver cell lines (Figure 2). Interestingly, this is also in accordance with findings from the in silico prediction which indicates that most of the tea phytoconstituents do not possess any hepatotoxicity (Table 3) or inhibit vital cytochrome enzymes (Table 4) of liver drug metabolism, despite belonging to different toxicity classes with variable lethal doses. With the overall biological activities of a plant product attributed to its constituent chemical compounds, synergistic or antagonist interactions between its phyto-ingredients could mask the potential toxic effects of some active components over others [30].

The potential gastrointestinal absorption or bioavailability of a drug candidate in eliciting physiological effects after oral ingestion can be determined by understanding its physiochemical properties. In this regard, some scientists have developed drug discovery guidelines that could help identify possible drug candidates. Some of these theoretical procedures for calculating the drug-likeness of these chemical compounds includes the Veber rule and Lipinski rules of five. While the former rule indicates that a drug with good bioavailability must not have a TPSA value greater than 140, the latter theorizes that such bioactive chemical must not violate more than one out of five rules which include: molar mass < 500 Dalton, LogP < 5, molar refractivity between 40–130, hydrogen acceptor group < 10, and hydrogen donor groups < 5 [31]. As most of the *C. gratissimus* components had less than two violations of these rules (Table 4), this study’s findings further corroborate the pharmacological benefits of this *C. gratissimus* tea for medicinal purposes.

## 4. Materials and Methods

### 4.1. Procurement of Herbal Tea Bags and Extraction

The tea bags of *Croton gratissimus* (Moologa), a traditional herbal tea, were procured from the Indigenous Knowledge System’s (IKS) laboratory, Department of Pharmacology, University of the Free State, Bloemfontein, South Africa. Nine of the tea bags (16 g) were infused in 150 mL of boiled distilled water for 10 min. It was continuously stirred and pressed against the walls of the beaker using a spatula for thorough extraction. The resulting infusion was transferred into a pre-weighed beaker and further concentrated at ˂50 °C in a water bath for several hours. After concentration, the beaker was weighed, and a sticky dark brown extract of 17.8% yield (2.8 g) was obtained. The concentrated sample was stored in air-tight glass vials until analysis.

A stock solution of 1 mg/mL of the hot water infusion was prepared using distilled water. Various working concentrations of 15, 30, 60, 120, and 240 μg/mL were then prepared for subsequent biochemical analyses. The same preparation was completed for the antioxidant standards, ascorbic acid and gallic acid.

### 4.2. Phytochemical Screening and Characterization

#### 4.2.1. Estimation of Total Phenolic Content

The total phenolic content of each extract was determined (as gallic acid equivalent) according to a previously established method [32] with slight modifications. In brief, 200 µL of the extract (240 µg/mL) was incubated with 1 mL of 10 times diluted Folin ciocalteau reagent and 800 µL of 0.7 M Na_2_CO_3_ for 30 min at room temperature. Thereafter, each concentration of samples and standard (gallic acid) were placed in a 96-well plate in triplicates. The absorbance was measured at 765 nm by utilizing a Multiskan GO plate reader (Thermo Scientific, Ratastie, Finland). A gallic acid standard curve was constructed, and total phenolic content was calculated and expressed as gallic acid (GAE) equivalents in milligrams per gram of sample dry weight.

#### 4.2.2. Estimation of Total Flavonoid Content

The total flavonoid content (equivalent to quercetin) of the extracts was determined according to an established colorimetric method [33] with little modification. Briefly, 200 µL of the extract was mixed with 150 µL of aluminum chloride (10%), 150 µL of potassium acetate (1M), and 2700 µL of distilled water. The mixture was vortexed for 10 s for proper mixture and incubated for 30 min at room temperature. After the incubation, various concentrations of the extract and standard (quercetin) were placed in 96-well plates in triplicates, and the absorbance was measured at 415 nm. The total flavonoid content was extrapolated from a quercetin standard curve and expressed as equivalents of quercetin (QE) in milligrams per gram of sample dry weight.

### 4.3. Characterization of C. gratissimus Phytoconstituents with High-Performance Liquid Chromatography-Mass Spectrophotometry (HPLC-MS)

This chemical analysis was carried out by a direct loop injection of the infusion sample (1 µL) into a Shimadzu Single Quadrupole LC-MS (Shimadzu Company, Kyoto, Japan) instrument, and fitted with a LC-2030 pump, deuterium (D2) lamp, and LC-2030/2040 photodiode array detector. The MS was operated in the electrospray ionization mode using the procedure in Ma et al. [34] with few modifications. In total, 0.1% formic acid in water was employed as mobile phase A whereas methanol:acetonitrile (1:1) constituted the mobile phases B. The pump was operated at 0.3 mL/min flow rate in a low-pressure gradient mode. The oven temperature was 40 °C with a maximum of 50 °C while the MS operation time was between 0.0–50.0 min. The MS scan speed was 5000 u/s and the result generated (raw data) between 100 m/z and 1000 m/z, which was saved in CDF format before import to the Mzmine software (version 2.9). The computer application was used to generate the masses of the developed peaks, while the chromatogram of the different scans was created with a 0.05 min time limit setting (minimum) and mass/charge tolerance of 0.05–5.0 ppm. After the peaks chromatograms were deconvoluted and de-isotoped, those with similar isotope patterns were grouped, and the resulting data were generated by setting the software analysis parameters as Olawale et al. [35] described previously. This was employed for compound identification via searching the Kegg internet database.

### 4.4. Antioxidant In-Vitro Screening of Herbal the Infusion

#### 4.4.1. Diphenyl-2-Picryl-Hydrazyl (DPPH) Radical Scavenging Activity

The DPPH activity of the extract was determined according to a previously established method [36]. Briefly, 500 µL of a 0.3 mM solution of DPPH in methanol was added to various concentrations (15–240 µg/mL) of 1 mL of the herbal extract. The solutions were vortexed and incubated in the dark for a period of 30 min at room temperature. Thereafter, a 200 µL aliquot of the respective sample and standards (gallic acid and ascorbic acid) was immediately transferred into a 96-well plate. The absorbance was measured at 517 nm against the blank, devoid of the free radical scavenger. The results were calculated using the expression below:% Inhibition = [(Absorbance of control − Absorbance of sample)/Absorbance of control] × 100

#### 4.4.2. Ferric Reducing Antioxidant Power (FRAP) Assay

FRAP activity was determined according to a previously established method [37]. In brief, 50 µL of 0.2 M sodium phosphate buffer (pH 6.6) and 1% potassium ferricyanide were incubated with 50 µL of the extract for a period of 30 min at 50 °C. After incubation, 50 µL of 10% trichloroacetic acid was used to acidify the reaction mixture. Thereafter, 50 µL of ferric chloride (0.1%) and distilled water were added to the acidified mixture. The absorbance was measured at 700 nm, and the results were calculated using the expression below:% Reducing power = (Absorbance of sample/Absorbance of Gallic acid) × 100

### 4.5. Cell Culture and Cytotoxicity Screening

#### 4.5.1. Cell Maintenance

The Chang liver cell line (American Type Culture Collection, Manassas, VA, USA) was cultured in Dulbecco’s Modified Eagles Medium (DMEM) high glucose (4.5 g/L), L-glutamine, and 25 mM HEPES (Gibco^®^ byLife Technologies, ThermoFisher Scientific, Johannesburg, South Africa) that was supplemented with 10% fetal bovine serum (FBS). Next, 9.4 cm^2^ tissue culture Petri dishes (SPL Life Sciences, Pochon, Republic of Korea) were used for the culturing of the cell lines. The sub-culturing procedure was carried out by aspirating the spent media, and the cells were washed twice with 5 mL PBS (7.2) (Gibco^®^ by Life Technologies, ThermoFisher Scientific, Johannesburg, South Africa). After washing, 500 microliters of 0.25% Trypsin-EDTA was added, and the cells were incubated for 15 min at 37 °C. Cell detachment was observed through an inverted microscope. After cell detachment, 2 mL of the complete medium was added, and the cell lines were sub-cultured by utilizing the sub-cultivation ratio of 1:4. All the cells were acquired from the American Type Culture Collection (Manassas, VA, USA). The cell cultures were cultured at 37 °C in a humidified atmosphere with 5 percent carbon dioxide (CO_2_) concentration.

#### 4.5.2. MTT Assay

In a 96-well plate, Chang liver cells were seeded at a cell density of 100 µL/well and incubated overnight at 37 °C in a humidified environment (5% CO_2_) for the cells to attach. Thereafter, cells were incubated for 48 h with different concentrations of *C. gratissimus* (15–240 µg/mL) at 37 °C, with 3 µg/mL of doxorubicin as the positive control. Additionally, 0.5% of Dimenthylsulfoxide (DMSO) was utilized as the control. The cytotoxicity of the cells was determined using MTT (3-(4,5-dimethylthiazol-2-yl)-2,5-diphenyltetrazolium bromide.

After the cells were treated, the media spent was discarded and 0.5 mg/mL MTT solution (100 µL/well) was added, and the cells were incubated at 37 °C for 2 h. After incubation, The MTT solution was then discarded and replaced with 100 μL of DMSO. The absorbance was measured at 550 nm using the Multiskan GO plate reader (Thermo Scientific, Ratastie, Finland).

### 4.6. Pre-Treatment and Induction of Oxidative Injury on Chang Liver Cells

This was completed according to a previous procedure [38]. The cells were seeded overnight at 37 °C in 6-well plates (TPP^®^, Merck, Rahway, NJ, USA) at 160,000 cells/well (2 mL/well). Thereafter, cells were pre-treated with *C. gratissimus* extract or the standard; ascorbic acid and gallic acid (15–240 µg/mL), and then incubated for 25 min at 37 °C. Then, 7 mM iron sulphate (600 µL) was included in each well before a further 30 min incubation at the same temperature. The media was discarded after incubation, and the cells were rinsed in PBS before they were harvested with trypsin and collected into PBS. Cells were vortexed for 1 min before centrifuging and centrifuged at 20,000× *g* for 10 min (4 °C). The supernatant was stored in 2mL Eppendorf tubes at −80 °C were used for subsequent analyses.

### 4.7. Determination of Reduced Glutathione (GSH) Level

GSH levels were estimated by using the Ellman’s spectrophotometric method [39] with little modification. Briefly, 200 Μl aliquot of the supernatant was deproteinized with 600 Μl of trichloroacetic acid (10%) by centrifuging for 5 min at 1000× *g*. Then, 200 μL supernatant of the deproteinized solution was then added to a 96-well plate already containing 500 μL of Ellman’s reagent. The mixture was incubated for 10 min at room temperature, and the absorbance was measured at 415 nm. The GSH level was estimated from a glutathione standard curve.

### 4.8. Determination of Superoxide Dismutase (SOD) Activity

SOD activity was determined in the cell by a modified method of Kakkar et al. [40]. In brief, 170 microliters of 0.1 mM DETAPAC solution and 15 microliters of the sample or standards were pipetted into a 96 well. Thereafter, 15 microliters of 1.6 mM 6-HD was added. The mixture was instantly mixed by tapping all four sides of the 96-well plate. Subsequently, the absorbance values were determined at 492 nm using the Multiskan GO plate reader (Thermo Scientific, Ratastie, Finland) for 5 min at 1 min intervals.

### 4.9. Determination of Catalase Activity

Catalase activity was determined according to the method of Hadwan and Abed [41]. Briefly, 100 microliter of hydrogen peroxide (65 uM) was incubated with 100 microliters of the sample in 6.0 mM sodium phosphate buffer (pH 7.4) for 2 min at 37 °C. The reaction was stopped by adding 0.1 mL of 32.4 mM ammonium molybdate. The absorbance was read at 347 nm against a blank, using the Multiskan GO spectrophotometer (Thermo Scientific, Ratastie, Finland).

### 4.10. Determination of Malondialdehyde (MDA) Levels

The lipid peroxidation level in the haptic cells was estimated by measuring the MDA equivalence in the cells using a previously established method [42]. A solution mixture made up of 100 μL of the supernatant, an equal volume of sodium dodecyl sulfate (SDS; 8.1%) solution, 375 μL of pure acetic acid (20%), and 1 mL of 0.25% thiobarbituric acid (TBA) was placed in a water bath and boiled for 60 min. After boiling and cooling, an aliquot of 250 μL was placed in a 96-well plate, and the absorbance was measured at 532 nm to estimate to the MDA levels present in the cells.

### 4.11. Determination of Nitric Oxide (NO) Level

Nitric oxide content of the cells was determined by using the Greiss method, as described by Tsikas [43]. A solution containing 0.1 mL of the supernatant and 0.1 mL of Greiss reagent was incubated in the dark for 0.5 h at room temperature. In place of the supernatant, distilled water represents the blank. After incubation, the absorbance of the solution was read at 548 nm.

### 4.12. Molecular Docking Analysis of C. gratissimus Phytochemicals with Enzymatic Antioxidant Protein Targets

This analysis was carried out using the computer algorithm of the Autodock Vina program of Chimera software (V.1.16). Firstly, the SDF format of the *C. gratissimus* metabolites’ 3D structures was retrieved from the PubChem database before their co-crystallized water molecules were removed using the Dock prep tool of the Chimera application. It further minimizes the compounds’ structure to their best energetically favorable conformation and adds polar hydrogen atoms. The 3D X-ray crystallographic structures of superoxide dismutase and catalase were retrieved from the protein data bank (PDB). The two proteins with 2C9V and IF4J access codes were selected based on their domain completeness and good resolutions of less than 2.5 Å. Then, the proteins were prepared using a similar tool employed for the plant phytochemicals before the location of their active site was identified using the CASTp online server. Subsequently, molecular docking of the compounds was conducted in a grid box created to cover the whole of the active site pocket. After recording the binding free energy for the best pose of each compound at the active site, BIOVIA Discovery Studio was employed to visualize the 2D images of the resulting protein–ligand complex.

### 4.13. Pharmacokinetics and Drug-Likeness Analysis of C. gratissimus Phytochemical Constituents

The adsorption, distribution, metabolism, as well as excretion pharmacokinetic properties of *C. gratissimus* HPLC-MS, identified constituents which were evaluated using the SwissADME online predictive program available at http://www.swissadme.ch/index.php (accessed on 22 April 2023) as described in previous studies [44,45]. Firstly, the canonical simplified molecular-input line-entry system (SMILE) of each of the plant compounds was obtained from PubChem and then entered onto the website in a list with their respective names. This then generated the compounds’ Lipinski, Veber, and lead likeness characteristics. Moreover, the online tool also predicted the degree of solubility (TPSA, iLOGP), gastrointestinal absorption, and potential adverse interactions of the phytochemicals with key cytochrome P450 proteins involved in drug metabolism (CYP1A2, CYP2D6, CYP3A4).

### 4.14. Toxicity Prediction of C. gratissimus Phytochemical Constituents

The probable toxicological endpoints of the chemical compounds present in the *C. gratissimus* plant material was estimated using the selected toxicity parameters of the Pro-Tox II online virtual laboratory (https://tox-new.charite.de/protox_II/ (accessed on 22 April 2023)). It calculated the lethal dose 50 (LD_50_) and toxicity class of the plant compounds while predicting the hepatic organ toxicity and carcinogenicity, as well as possible activation of the estrogen receptor-α (ER) and nuclear factor (erythroid-derived 2)-like 2/antioxidant responsive element (nrf2/ARE) stress pathway signaling.

### 4.15. Statistical Analysis of Data

Data were presented as mean ± SD, and experiments were carried out in triplicates (n = 3). Data analysis was completed using SPSS (Windows V25). Statistically significant differences (*p* < 0.05) between test groups were established using a one-way analysis of variance and followed by the comparison of mean values with Tukey’s HSD multiple range test.

## 5. Conclusions

Given that oxidative stress plays a critical role in the development and progression of several liver diseases, it is crucial to continue exploring natural phytoconstituents of medicinal plants that possess strong antioxidant effectiveness, which are safer and effective towards the treatment and management of liver diseases. The results from the present study suggest the possible hepatoprotective potential of *C. gratissimus* herbal tea against iron-induced oxidative hepatic injury. This is due to the DPPH scavenging and ferric reducing capacity of the herbal extract as well as its ability to mitigate oxidative stress in iron-induced hepatic injury. These observed activities may be attributed to the synergistic effect of identified phytochemicals of the herbal tea. In silico studies of the tea’s phytochemicals revealed a strong interaction with the studied antioxidant enzymes and they exhibited a promising pharmacological profile and bioavailability. However, further in vivo and molecular studies are required to ascertain these results and understand the mechanisms by which *C. gratissimus* exhibits its hepatoprotective effects. The limitation of the present study is the lack of quantitative analysis of the identified tea compounds. Therefore, we propose future quantitative analysis and possible compound isolation studies to characterize the *C. gratissimus* bioactive phytoconstituents.

## Figures and Tables

**Figure 1 plants-12-02915-f001:**
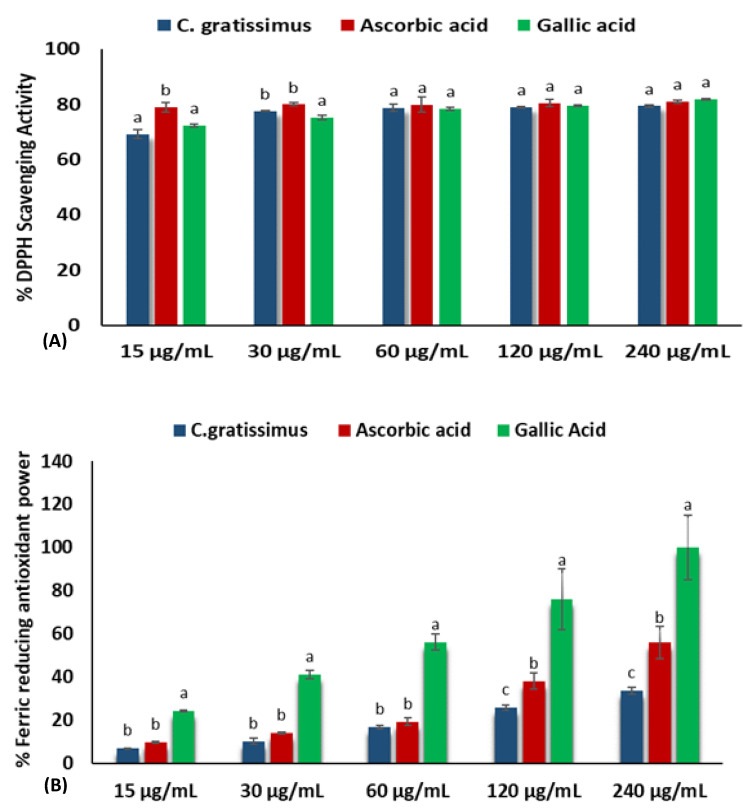
(**A**) DPPH scavenging and (**B**) FRAP activities of *C. gratissimus*. Data is presented as mean ± SD. Each of the alphabetical letters (^a–c^) above the bars for a given concentration illustrate the statistical significance of difference (*p* < 0.05), Tukey’s-HSD multiple range post hoc test.

**Figure 2 plants-12-02915-f002:**
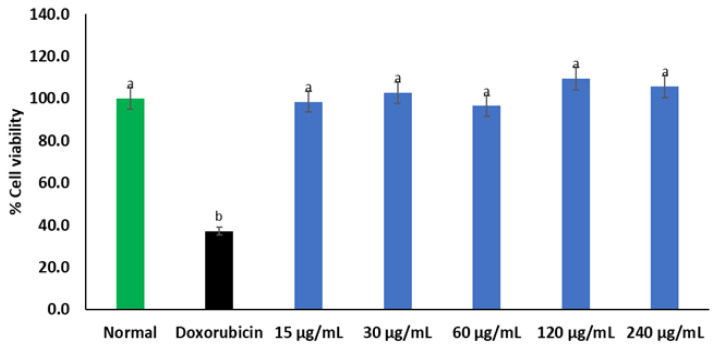
Cytotoxic effect of *C. gratissimus* on Chang liver cell line. Data presented as mean ± standard deviation. ^a^ Statistically significant compared to doxorubicin group; ^b^ statistically significant compared to the normal control cell (*p* < 0.05, Tukey’s HSD-multiple range post hoc test).

**Figure 3 plants-12-02915-f003:**
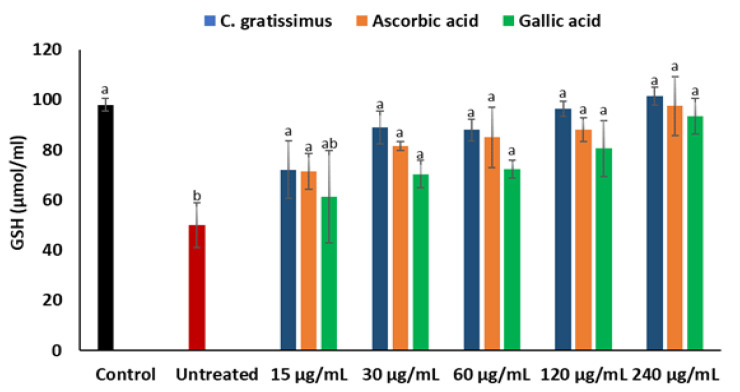
Effect of *C. gratissimus* on GSH concentration in oxidative hepatic injury. Value = mean ± *SD*; *n* = 3. ^a^ Statistically significant compared to untreated hepatic cells; ^b^ statistically significant compared to the control cells; ^ab^ statistically significant compared to both control and untreated cells (*p* < 0.05, Tukey’s HSD-multiple range post hoc test). Control = Normal hepatic cells; Untreated = Oxidative injured cells not treated.

**Figure 4 plants-12-02915-f004:**
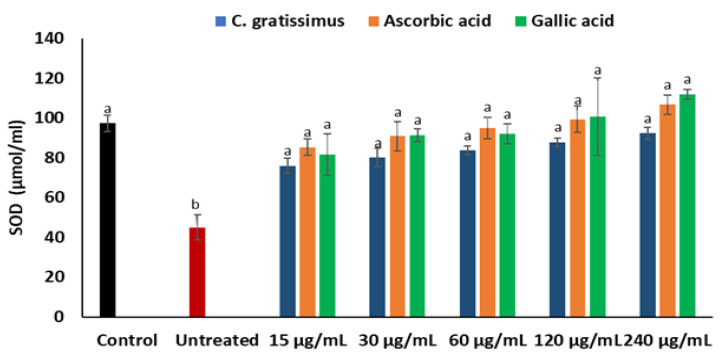
Effect of *C. gratissimus* on SOD activity in oxidative hepatic injury. Value = mean ± *SD*; *n* = 3. ^a^ Statistically significant compared to untreated hepatic cells; ^b^ statistically significant compared to the control cells; ^ab^ statistically significant compared to both control and untreated cells (*p* < 0.05, Tukey’s HSD-multiple range post hoc test). Control = Normal hepatic cells; Untreated = Oxidative injured cells not treated.

**Figure 5 plants-12-02915-f005:**
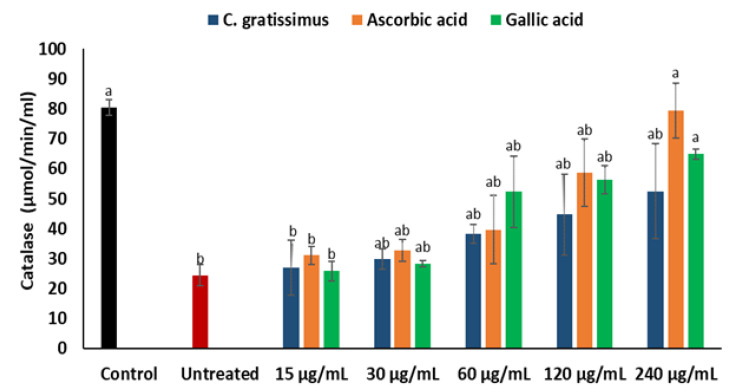
Effect of *C. gratissimus* on catalase activity in oxidative hepatic injury. Value = mean ± *SD*; *n* = 3. ^a^ Statistically significant compared to untreated hepatic cells; ^b^ statistically significant compared to the control cells; ^ab^ statistically significant compared to both control and untreated cells (*p* < 0.05, Tukey’s HSD-multiple range post hoc test). Control = Normal hepatic cells; Untreated = Oxidative injured cells not treated.

**Figure 6 plants-12-02915-f006:**
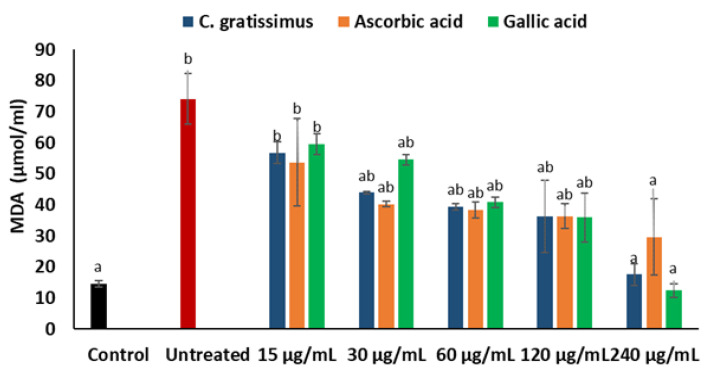
Effect of *C. gratissimus* on malondialdehyde (MDA) levels in oxidative hepatic injury. Value = mean ± *SD*; *n* = 3. ^a^ Statistically significant compared to untreated hepatic cells; ^b^ statistically significant compared to the control cells; ^ab^ statistically significant compared to both control and untreated cells (*p* < 0.05, Tukey’s HSD-multiple range post hoc test). Control = Normal hepatic cells; Untreated = Oxidative injured cells not treated.

**Figure 7 plants-12-02915-f007:**
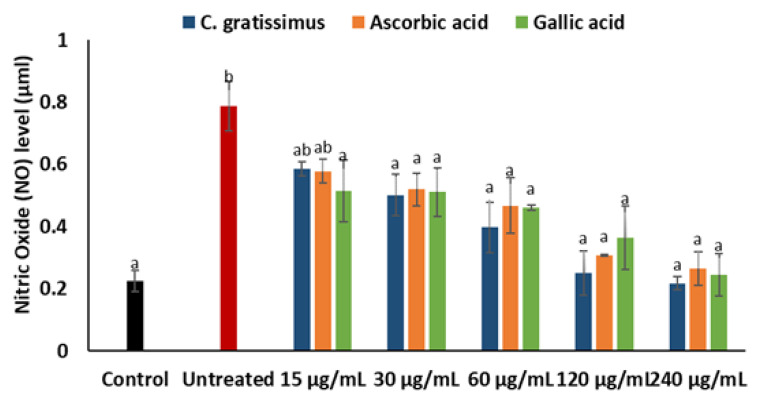
Effect of *C. gratissimus* on nitric oxide (NO) levels in oxidative hepatic injury. Value = mean ± *SD*; *n* = 3. ^a^ Statistically significant compared to untreated hepatic cells; ^b^ statistically significant compared to the control cells; ^ab^ statistically significant compared to both control and untreated cells (*p* < 0.05, Tukey’s HSD-multiple range post hoc test). Control = Normal hepatic cells; Untreated = Oxidative injured cells not treated.

**Figure 8 plants-12-02915-f008:**
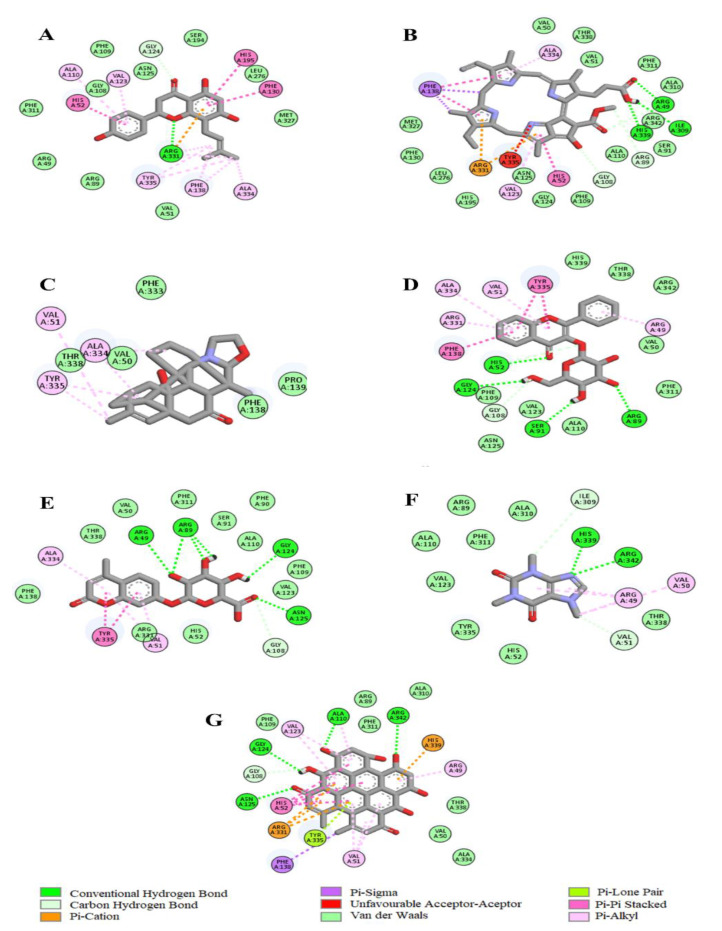
2D image virtualization of the catalase enzyme active site molecular interactions with (**A**) 8-Prenylnaringenin, (**B**) Pheophorbide a, (**C**) Spirasine I, (**D**) Flavonol 3-O-D-galactoside, (**E**) 4-Methylumbelliferone glucuronide, (**F**) Caffeine, and (**G**) Hypericin.

**Figure 9 plants-12-02915-f009:**
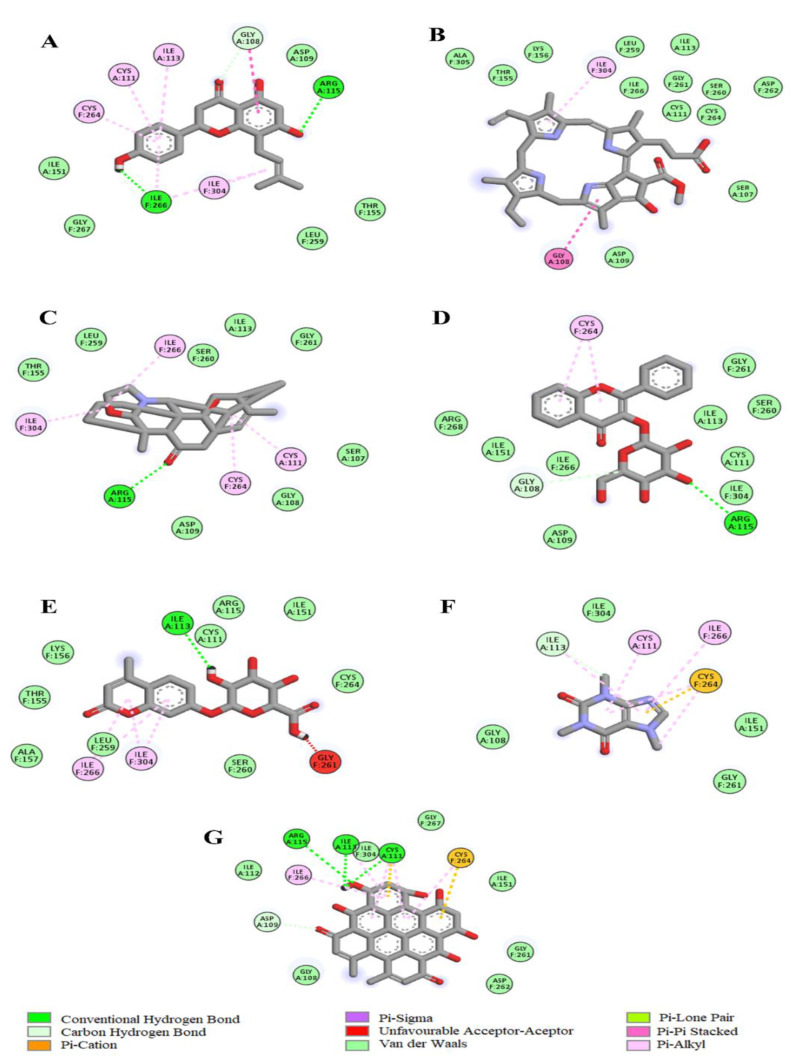
2D image virtualization of the superoxide dismutate enzyme active site molecular interactions with (**A**) 8-Prenylnaringenin, (**B**) Pheophorbide a, (**C**) Spirasine I, (**D**) Flavonol 3-O-D-galactoside, (**E**) 4-Methylumbelliferone glucuronide, (**F**) Caffeine, and (**G**) Hypericin.

**Table 1 plants-12-02915-t001:** IC_50_ values of *C. gratissimus* infusion on different assayed parameters.

Antioxidative Activity	*C. gratissimus*	Ascorbic Acid	Gallic Acid
DPPH	0.01	0.01	0.02
FRAP	>1000	229.82	42.12

Values are expressed as µg/mL. DPPH: 1-Diphenyl-2-Picryl-Hydrazyl radical scavenging activity, FRAP: free radical antioxidant power.

**Table 2 plants-12-02915-t002:** *C. gratissimus* herbal tea infusion identified secondary metabolites.

Compounds	[M + H] m/z	Ion Intensity (Area)	Molecular Weight	Molecular Formular
8-Prenylnaringenin	341.1	6,745,309.5	340.3	C_20_H_20_O_5_
Pheophorbide a	593.2	131,495.5	592.6	C_35_H_36_N_4_O_5_
Spirasine I	355.4	1,006,042.8	355.4	C_22_H_29_NO_3_
Flavonol 3-O-D-galactoside	401.1	941,414.5	400.3	C_21_H_20_O_8_
4-Methylumbelliferone glucuronide	353.0	117,078,415.1	352.3	C_16_H_16_O_9_
Caffeine	195.1	21,930,284.6	195.1	C_8_H_10_N_4_O_2_
Hypericin	505.1	3,182,449.0	504.4	C_30_H_16_O_8_

**Table 3 plants-12-02915-t003:** Potential oral toxicity and molecular binding free energies (Kcal/mol) of *C. gratissimus* phytochemicals with enzymatic antioxidants.

	LD_50_ (mg/kg)	Toxicity Class	Hepatotoxicity	Nrf/ARE	Catalase	SOD
8-Prenylnaringenin	2000	4	No	No	−10.6	−6.6
Pheophorbide-a	40	2	No	No	−12.6	−4.7
Spirasine I	260	3	No	No	−6.0	−7.0
Flavonol 3-O-D-galactoside	5000	5	No	No	−9.0	−6.6
4-Methylumbelliferone glucuronide	5000	5	No	No	−10.3	−7.6
Caffeine	127	3	No	No	−7.5	−4.6
Hypericin	1000	4	No	No	−10.3	−6.7

LD_50_ = Lethal dose 50; Nrf2/ARE = Nuclear factor (erythroid-derived 2)-like 2/antioxidant responsive element; SOD = Superoxide dismutase.

**Table 4 plants-12-02915-t004:** Druglikeness and pharmacokinetic characteristics of *C. gratissimus* phytochemical components.

	TPSA	LOGP	GIA	CYP1A2 Inhibitor	CYP2D6 Inhibitor	CYP3A4 Inhibitor	No of Lipinski Violations	No of Veber Violations	No of PAINS Alerts	No of Leadlikeness Violations
8-Prenylnaringenin	86.99	2.59	High	Yes	Yes	Yes	0	0	0	1
Pheophorbide-a	132.94	3.31	Low	No	No	No	1	0	0	1
Spirasine I	49.77	2.96	High	No	No	No	0	0	0	1
Flavonol 3-O-D-galactoside	129.59	1.71	High	No	No	No	0	0	0	1
4-Methylumbelliferone glucuronide	146.66	1.04	Low	No	No	No	0	1	0	1
Caffeine	61.82	1.79	High	No	No	No	0	0	0	1
Hypericin	155.52	3.10	Low	No	No	No	2	1	1	2

TPSA = Topological polar surface area; LOGP = Lipophilicity; CYP = Cytochrome; Pan-assay interference compounds = PAINS; GIA = Gastrointestinal absorption.

## Data Availability

Not applicable.

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
