# Peer review of "Phytochemical Properties of Croton gratissimus Burch (Lavender Croton) Herbal Tea and Its Protective Effect against Iron-Induced Oxidative Hepatic Injury"

_plants, 2023, doi:10.3390/plants12162915_

Round 1

Reviewer 1 Report

The title of manuscript is remarkable. English language has good quality. Figures and tables have acceptable quality. There are some explainations that are needed about the section "Results" and "Discussions"

1.  About Figure 2

+ why do you think doxorubicin had significantly reduced viability?

+ Why the cell viability at the concentration of 120 ug/ml is more than that one in 30ug/ml?

2. About Figure 3

Why the level of GSH for cells treated with C. gratissimus had a little deduction in the concentration of 60ug/ml in comparison to those treated with 30ug/ml?

3. The authors should talk about future insights, obstackles and limittations that

related to their work in a separate section in the part "Results"

4. The authors should discuss about their bioavailability analysis in a separate part in the part "Discussion" and compare their results with other similar compounds and drugs

5. Why the authors have not discuss about the future of compounds that were undergone bioavailability prediction (compounds in table 4) as curative compounds?

6. Why the authors have not mentioned any thing about Compound accessibility and drug likeness score in table 4?

7. Please check and adjust the "Reference list" based on the regulations of reference list of journal. (Titles, doi, the name of journal

and ... )

Author Response

Response to reviewers’ comments

Manuscript ID: plants-2548918

Comments from reviewers:

Reviewer 1

The title of manuscript is remarkable. English language has good quality. Figures and tables have acceptable quality. There are some explainations that are needed about the section "Results" and "Discussions"

Response:

Thank you for taking time to review our manuscript. We believe your suggestions will improve our manuscript significantly. We have corrected the entire manuscript as per your suggestions and our responses in this regard are given below.

Comment 1:

 About Figure 2

+ why do you think doxorubicin had significantly reduced viability?

Response:

Doxorubicin is a clinically approved cytotoxic compound for cancer treatment. It is medically employed for limiting cancer cell proliferation via inhibition of the topoisomerase enzyme. Consequently, the lower cell viability of standard drug treatment group could be attributed to its possible cytotoxic effect on the normal liver cells.

+Why the cell viability at the concentration of 120 ug/ml is more than that one in 30ug/ml?

Response:

Although our result in Figure 2 suggests that there was no statistically significant difference between the cell viability at 120 ug/ml and 30 ug/ml, a higher viability at the initial concentration may be linked to potential increased protective effect of the tea extract on growth and multiplication of the cells. 

 Comment 2:

About Figure 3

Why the level of GSH for cells treated with C. gratissimus had a little deduction in the concentration of 60ug/ml in comparison to those treated with 30ug/ml?

Response:

Results in Figure 3 suggest that C. gratissimus appeared to exert a dose-dependent effect on GSH concentration. However, the slight reduction in the protein concentration at 60 ug/ml may be linked to variation due to pipetting. Despite this outcome, statistical analysis revealed that there are no differences between the treatments at the (60ug/ml) concentration.

Comment 3:

The authors should talk about future insights, obstackles and limittations that

 related to their work in a separate section in the part "Results"

Response:

This has been included in the result section. Please see lines 158-160

Comment 4:

The authors should discuss about their bioavailability analysis in a separate part in the part "Discussion" and compare their results with other similar compounds and drugs

Response:

This present study is focused mainly on investigating the potential protective effects of C. gratissimus tea on redox imbalance in the liver. In this regard, more studies are still required to confirm these phytochemicals. These studies will be accompanied by detailed description of all the pharmacokinetic properties, including the bioavailability of these phytoconstituents. Nonetheless, a session of this manuscript in lines 287 – 298 discussed the bioavailability of the tentative compound identified from this tea. Moreover, information in lines 185 -193 also mentioned other properties of the plant compounds, as displayed in Table 4

Comment 5:

Why the authors have not discuss about the future of compounds that were undergone bioavailability prediction (compounds in table 4) as curative compounds?

Response:

This study is not focused on identifying curative bioactive compounds from C. gratissimus tea. Instead, it seeks to determine the antioxidative effects of the plant in vitro while tentatively characterizing possible compounds in the tea that could be linked to the observed biological activity. However, experiments involving bioactivity-guided isolation of curative agents from the plant will be considered in future studies.

Comment 6:

Why the authors have not mentioned any thing about Compound accessibility and drug likeness score in table 4?

Response:

Though we acknowledge that there are different pharmacokinetic properties described in Table 4. However, most of these have been discussed in lines 185 -193 and 287 -298.

Comment 7:

Please check and adjust the "Reference list" based on the regulations of reference list of journal. (Titles, doi, the name of journal and ... )s

Response:

All the reference list has been adjusted accordingly and their respective DOI has been added.

Reviewer 2 Report

The present work reports the phytochemical properties of Croton gratissimus Burch herbal tea and its protective effect against iron-induced oxidative hepatic injury, using an in vitro and in silico approach. C. gratissimus herbal infusion was screened for total phenolic and total flavonoid contents as well as in vitro antioxidant capacity using both FRAP and DPPH methods. Oxidative hepatic injury was induced by incubating 0.007 M FeSO4 with Chang liver cells, initially incubated with or without different concentrations, of C. gratissimus infusion or the standard antioxidants, namely, gallic acid and ascorbic acid. Further, LC-MS analysis of the herbal tea revealed the presence of seven bioactive compounds.

The work is interesting and conducted with adequate means. The results suggest the hepatoprotective and antioxidative potentials of C. gratissimus herbal tea against oxidative hepatic injury.

- English requires a moderate revision.

- Line 340, I am not sure about the model of the LC-MS system. Also, a full description of the instrument must be provided.

- Line 343, Remove the comma after Ma.

-Lines 343-344, It is not clear the composition of the two mobile phase (A and B) used for the analysis.

Table 2. C. gratissimus herbal tea infusion identified secondary metabolites. Only qualitative data are reported. In the experimental section no reference material is reported. How was identification carried out? I invite the authors to provide quantification data to support their findings.

Moderate revision

Author Response

Response to reviewers’ comments

Manuscript ID: plants-2548918

Comments from reviewers:

Reviewer 2

The present work reports the phytochemical properties of Croton gratissimus Burch herbal tea and its protective effect against iron-induced oxidative hepatic injury, using an in vitro and in silico approach. C. gratissimus herbal infusion was screened for total phenolic and total flavonoid contents as well as in vitro antioxidant capacity using both FRAP and DPPH methods. Oxidative hepatic injury was induced by incubating 0.007 M FeSO4 with Chang liver cells, initially incubated with or without different concentrations, of C. gratissimus infusion or the standard antioxidants, namely, gallic acid and ascorbic acid. Further, LC-MS analysis of the herbal tea revealed the presence of seven bioactive compounds. The work is interesting and conducted with adequate means. The results suggest the hepatoprotective and antioxidative potentials of C. gratissimus herbal tea against oxidative hepatic injury.

Response:

Thank you for taking time to review our manuscript. We believe your suggestions will improve our manuscript significantly. We have corrected the entire manuscript as per your suggestions and our responses in this regard are given below.

Comment 1:

English requires a moderate revision.

Response:

The whole manuscript has been corrected for grammatical errors.

Comment 2:

Line 340, I am not sure about the model of the LC-MS system. Also, a full description of the instrument must be provided.

Response:

Thank you for your observation, the LC-MS model utilized, with detailed operational condition recorded during the LC-MS analysis have been provided. Please see lines 336-346

Comment 3:

Line 343, Remove the comma after Ma.

Response:

It has been removed.

Comment 4:

Lines 343-344, It is not clear the composition of the two mobile phase (A and B) used for the analysis.

Response:

In this study’s LC-MS analysis, two mobile phases were used. These include 0.1% formic acid in water which is mobile phase A and methanol:acetonitrile (1:1) as mobile phase B,  as indicated in lines 340-341

Comment 5:

Table 2. C. gratissimus herbal tea infusion identified secondary metabolites. Only qualitative data are reported. In the experimental section no reference material is reported. How was identification carried out? I invite the authors to provide quantification data to support their findings.

Response:

The LC-MS characterization carried out in this study was done to identify possible phytochemicals in the tea that may be linked with the activities observed. The raw data generated from the LC-MS analysis was exported to MZmine software which then present the probable compounds present in the sample via comparing the m/z fragments with those of other compounds in the Kegg database. Interestingly, this software has been used for compound identification in other published works as those below;

  • D’Urso, G., Pizza, C., Piacente, S., & Montoro, P. (2018). Combination of LC–MS based metabolomics and antioxidant activity for evaluation of bioactive compounds in Fragaria vesca leaves from Italy. Journal of pharmaceutical and biomedical analysis, 150, 233-240.
  • Michl, J., Bello, O., Kite, G. C., Simmonds, M. S., & Heinrich, M. (2017). Medicinally used Asarum species: high-resolution LC-MS analysis of aristolochic acid analogs and in vitro toxicity screening in HK-2 cells. Frontiers in pharmacology, 8, 215.

However, quantifying these metabolites in the test sample will require characterizing them with known standards compounds. Since this study identified these constituents in the tea sample, not quantifying them may be the limitation of the present study. Future work has thus been proposed to confirm their presence via compound isolation studies (kindly see lines 499-502).

Reviewer 3 Report

1- Please add the English name of Croton gratissimus in the title and in the manuscript (both abstract and introduction).

2- Arrange keywords on the basis of alphabetical order and the first letter each word in keywords should be small.

3- There are many paragraphs in the introduction and in the manuscript, each paragraph should start with new findings and new points and manage paragraphing in the manuscript.

4- Conclusion should be revised and re-written completely. The manuscript also needs at least one paragraph which give introductions to the readers and scientists for future researches and directions. Please, add this part as well as improve Conclusion part.

5- Some references are not on the basis of journal s format. Please, check reference 45, the name of journal should be italics.

6- All articles needs DOI. Please, add DOI for all articles in Reference part.

Minor English editing is needed for the manuscript.

Author Response

Response to reviewers’ comments

Manuscript ID: plants-2548918

Comments from reviewers:

Reviewer 3

Comments and Suggestions for Authors

Thank you for taking time to review our manuscript. We believe your suggestions will improve our manuscript significantly. We have corrected the entire manuscript as per your suggestions and our responses in this regard are given below.

Comment 1:

Please add the English name of Croton gratissimus in the title and in the manuscript (both abstract and introduction).

Response:

Thank you for your suggestion. It has been included.

Comment 2:

Arrange keywords on the basis of alphabetical order and the first letter each word in keywords should be small.

Response:

It has been corrected.

Comment 3:

There are many paragraphs in the introduction and in the manuscript, each paragraph should start with new findings and new points and manage paragraphing in the manuscript.

Response:

Thank you for your observation, the manuscript paragraphing has been improved.

Comment 4:

Conclusion should be revised and re-written completely. The manuscript also needs at least one paragraph which give introductions to the readers and scientists for future researches and directions. Please, add this part as well as improve Conclusion part.

Response:

The conclusion section has been completely revised accordingly. Kindly see lines 487-502

Comment 5:

Some references are not on the basis of journal s format. Please, check reference 45, the name of journal should be italics.

Response:

All the reference list has been corrected accordingly.

Comment 6:

All articles needs DOI. Please, add DOI for all articles in Reference part.

Response:

Respective DOI has been added to the references.

Round 2

Reviewer 1 Report

There is no more suggestion. Thanks

Reviewer 2 Report

The authors have adequately addressed all Reviewers remarks. The paper can be now accepted in the present form.